# Experimental study protocol of the project "MOtor function and VItamin D: Toolkit for motor performance and risk Assessment (MOVIDA)"

**Valeria Belluscio**[1☺], **Amaranta S. Orejel Bustos**[1☺], **Valentina Camomilla**[1], **Francesco Rizzo**[2], **Tommaso Sciarra**[2], **Marco Gabbianelli**[3], **Raffaella Guerriero**[3], **Ornella Morsilli**[3], **Francesco Martelli**[3], **Claudia Giacomozzi**[3]*

1 Interuniversity Centre of Bioengineering of the Human Neuromusculoskeletal System (BOHNES), Department of Movement, Human and Health Sciences, University of Rome "Foro Italico", Rome, Italy, 2 Joint Veterans Defence Center, Army Medical Center, Rome, Italy, 3 Department of Cardiovascular and Endocrine-Metabolic Diseases and Aging, Italian National Institute of Health, Rome, Italy

☺ These authors contributed equally to this work.
* claudia.giacomozzi@iss.it

**Data Availability Statement:** No datasets were generated or analysed during the current study. All relevant data from this study will be made available

## Abstract

Musculoskeletal injuries, a public health priority also in the military context, are ascribed to several risk factors, including: increased reaction forces; low/reduced muscle strength, endurance, body mass, Vitamin D level, and bone density; inadequate lifestyles and environment. The MOVIDA Project–funded by the Italian Ministry of Defence—aims at developing a transportable toolkit (assessment instrumentation, assessment protocols and reference/risk thresholds) which integrates motor function assessment with biological, environmental and behavioural factors to help characterizing the risk of stress fracture, stress injury or muscle fatigue due to mechanical overload. The MOVIDA study has been designed following the STROBE guidelines for observational cross-sectional studies addressing healthy adults, both militaries and civilians, with varying levels of physical fitness (sedentary people, recreational athletes, and competitive athletes). The protocol of the study has been designed and validated and is hereby reported. It allows to collect and analyse anamnestic, diagnostic and lifestyle-related data, environmental parameters, and functional parameters measured through portable and wearable instrumentation during adapted 6 minutes walking test. The t-test, one and two-way ANOVA with post-hoc corrections, and ANCOVA tests will be used to investigate relevant differences among the groups with respect to biomechanical parameters; non-parametric statistics will be rather used for non-normal continuous variables and for quantitative discrete variables. Generalized linear models will be used to account for risk and confounding factors.

upon study completion if authorized by Data Protection Officer of involved Institutions.

**Funding:** This research was funded by the Italian Ministry of Defence, grant "P.N.R.M. 2019 Capitolo 7101/01. Ricerca MOVIDA".

**Competing interests:** The authors have declared that no competing interests exist.

## Introduction

### Background

Overuse injuries (bone fractures, musculoskeletal injuries), occurring when involved structures fail to adapt to increased mechanical stress [1–5], mostly affect resistance athletes, athletes involved in pre-season preparation, and military populations [3, 5–15]. In the army, most injuries occur both during the training of new recruits and in special training contexts (incidence up to ≈7% for men and 21% for women [16]) [3, 10, 11, 13, 15–17]. The implications in terms of patient morbidity, recurrence, reduced military performance, and dropout rates do represent a relevant public health problem and a demand for efficient surveillance, prevention and treatment plans [3, 11, 15].

Among the several suggested risk factors, worth to mention are: increased ground reaction forces during motion tasks; low/reduced muscle strength, body mass index [3], bone mineral density and Vitamin D level reduction [7, 18, 19]; lifestyle factors; footwear [3]; training and training environment variables [3]; age; sex; ethnicity; previous overuse injuries; lower limbs altered morphology and function; metabolic alterations [4, 5, 8, 10–14, 17, 20, 21]. With reference to Vitamin D, the high incidence of D hypovitaminosis in the global population is currently of remarkable interest [22–25], also because it has been associated with higher percentages of stress fractures [18, 19], and it has been found even in young people [9, 26, 27]. Moreover, in the military populations, there is a significant predisposition to develop low levels of Vitamin D or to undergo relevant level fluctuations [7, 9], with high D hypovitaminosis incidence and prevalence among soldiers [new28–new30]. This phenomenon has been partly ascribed to low exposure to sunlight [22] and to the lack of adequately supplemented food [7]. Finally, possible mutual interferences between Vitamin D level and physical training still need to be clarified [7, 19, 31, 32], either when the latter is excessive or even when it is too poor or completely missing due to reduced mobility, as in the case of veterans.

Besides soldiers, many of the above risk factors do affect several populations [19, 33], thus representing a research and intervention priority for healthcare systems [3]. Among these populations, great attention is paid to the elderly, where many of the mentioned risk factors are found simultaneously [24, 34] and "overuse" mostly results from the application of even small mechanical load to fragile structures, and to competitive athletes whose musculoskeletal structures, though well trained, are frequently exposed to overuse especially in view of competitions. Recreational athletes recovering from injuries may similarly undergo to overuse-related injuries [3] and are thus also worth of accurate assessment.

Within the MOVIDA Project (MOtor functional and VItamin D: toolkit for motor performance and risk Assessment (MOVIDA); Oct 31st 2019 –January 28th 2021; P.N.R.M. Capitolo 7101/01. Ricerca MOVIDA), approved and funded by the Italian Ministry of Defence, a transportable toolkit (instrumentation, web-based repository, protocols and reference/risk thresholds) has been proposed; the toolkit integrates motor function with biological, clinical, environmental and behavioural factors to help characterizing the risk of overuse-related musculoskeletal injury. A predictive model, relying on the toolkit and on the effective characterization of musculoskeletal overuse injury risk exposure in a broad healthy population including militaries, civilians and athletes, might represent a valuable support for strategies like monitoring, prevention and rehabilitation treatment of overuse musculoskeletal injuries [3, 35, 36]. Once fully validated, the model might be effectively translated into the clinical practice to integrate care models of fragile, risk-exposed populations [37].

The funded MOVIDA Project relied on two relevant items, namely the setup and validation of the toolkit and the use of the toolkit within the project associated Observational Cross-Sectional Study (hereby referred to as OCSS MOVIDA), so as to complete the model.

The first item has been fully completed within the time-frame of the project, also relying on valuable literature [38–49] retrieved and synthesized within the MOVIDA Project systematic review [new ref 50]. Due to limitations related to Covid-19 emergency, the data collection to address the second item has not started yet. Despite the project has officially concluded at the end of January, authorizations and agreements have been obtained to perform the study beyond the end of the project, with a possible start in-between March and June 2021. In the following, the protocol of OCSS MOVIDA, designed on the basis of STROBE Guidelines [51] is carefully described by using the STROBE checklist of items to be included in reports of cross-sectional studies [29, 52].

## Objectives

The main objective of the OCSS MOVIDA is to get the characterization, as complete as possible, of musculoskeletal overuse injury risk exposure in a broad military and civilian healthy population by exploiting the portable toolkit setup (validated as first item of the MOVIDA Project).

Specific objectives of the study are: i) the analytical observation of exposed populations (also including the setup of a database to be fed); within the MOVIDA Project, samples of specific interest were defined as military and civilian adults, grouped with respect to the intensity of physical training into a sedentary group, a recreational athletes group, and an elite athletes group; those samples should be characterized with respect to the risk of musculoskeletal overuse injuries, in relation to training, clinical history, biological characteristics, lifestyles, and biomechanical outcomes from the functional tests; ii) modelling of risk factors interaction, including modelling to associate the assessment outcomes with both retrospective (previous injuries occurred within 2 years) and prospective (during a follow-up period up to 2 years) information; in particular, the Vitamin D biomarker will be monitored within the examined cohort to possibly include it as a relevant co-variate in the models; and iii) optimization for the applicability of the assessment protocol to other exposed populations, especially to the elderly.

## Methods and analysis

### Study design

Key elements of the study design, detailed in the following paragraphs, can be summarized as follows:

A. the study population: definition and recruitment;

B. the collection of anamnestic and lifestyle data through an online platform;

C. the instrumental functional assessment;

D. the acquisition of specific clinical-diagnostic biomarkers;

E. the tools and methods of analysis and synthesis of experimental data;

F. the interventions to contain the risk of contagion from Covid-19.

The key-element F, a mandatory requirement due to the still ongoing worldwide emergency due to Covid-19, has been addressed at different levels in all the above elements, as specified in the below detailed descriptions. The explanations of key-elements from B to D have been structured in terms of instrumentation or tools, protocols, and outcomes.

## Setting

In the initial GANTT chart of the MOVIDA Project, the conduction of the study had been planned from the fourth to the ninth month of the project (namely from March 2020 to August 2020). Unfortunately, the experimental study was suspended, due to Covid-19 emergency, total or partial lockdowns on the Italian territory, restrictions imposed in working places and laboratories, and change of priorities in healthcare involved structures. At present, it is reasonable to hypothesize it will start between March and June 2021, with a possible duration of 9 rather than 6 months due to expected longer time required for both recruitment and study execution.

Different settings and locations were established for the three data retrieval steps corresponding to key-elements B to D of the study design:

- the collection of anamnestic and lifestyle data (key-element B) will be performed through a dedicated online platform built in the Moodle environment and hosted by the Italian National Institute (ISS) IT Service. Compilation of questionnaires and upload of possible supplemental documentation is expected to take 1 hour at maximum, with the "save and come back later" option. The only requirement is that data collection shall be completed at least one week before the functional and clinical-diagnostic examination;

- the instrumental functional assessment (key-element C) will take place at the ISS Authors' Lab of Motion Analysis (Rome, Italy), in collaboration with the Authors from the Foro Italico University. The whole assessment, including Covid-related and entrance authorization procedures, is expected to last approximately 1.5 hours;

- the acquisition of specific clinical-diagnostic biomarkers (key-element D) will occur at the Authors' Diagnostic Lab at the Celio Army Medical Center (Rome, Italy). The clinical-diagnostic examination is expected to last approximately 1 hour, including Covid-related and entrance authorization procedures.

The ISS and Celio premises are both located in the same area in Rome, at a distance of about 5 km. Whenever possible, participants will undergo the two examinations in the same day. Alternatively, the time-span between the two examinations has been fixed to 2 weeks at maximum.

To address the specific objective of modelling the association of the experimental assessment outcomes with prospective information, participants will be asked to report about any overuse-related injury possibly occurred during a follow-up period up to 2 years.

## Participants

Inclusion criteria: age 18–40 years; body mass 50–110 kg; height 1.5–2.0 m; shoe size 36–45 Italian size (5–11 U.S. size); stability, as for type and intensity, in practicing physical activity: individuals will be classified as elite (>6 hours of physical activity per week), recreational (2.5–6 hours of physical activity per week) or sedentary (<2.5 hour of physical activity per week), as defined by the WHO [53]; availability to move to Rome for the examinations.

Exclusion criteria: type I Diabetes; musculoskeletal injuries within the last 3 months; invalidating complications due to previous or current diseases (Covid19-related pathologies included); use of any walking aid; abrupt changes in the amount of weekly physical activity within the last three months.

Covid19-related clinical status (suffering from, healed from), if not associated with long-term invalidating complications, will be asked as well and accounted in the analysis; however, it will not represent an a-priori exclusion criterion.

For the enrolment of volunteers, a digital leaflet has been prepared, containing a summary of the objectives and methods of the study and the inclusion criteria. The leaflet will be

disseminated through the main social platforms on the web and the official web channels of the Authors' Institutions. Interested volunteers will contact the MOVIDA Principal Investigator for a pre-screening interview and to eventually give their "intention to participate".

As anticipated above, the participants will be allocated in three different groups, of 50 participants each, namely elite athletes, recreational athletes and sedentary people. The following rules have been defined to form the groups:

- a "first in" criterion will be applied to all participants, based on the date of agreed adhesion;

- groups should be as much homogeneous as possible with respect to the Women-To-Men ratio (WTM±0.2);

- to comply with the previous rule, the elite athlete group will be taken as reference for the WTM ratio;

- if necessary, the other two groups will be adjusted by randomly removing one participant of the sex which needs sample reduction and by replacing her/him with the first of the opposite sex in the waiting list; those steps may be eventually iterated;

- those excluded from the enrolment will be however offered to be examined afterwards; their availability will also be explored to be enrolled in case of drop-outs during the study.

## Variables, data sources and measurement

Variables are hereby listed according to the three data collection contexts associated with key-elements B, C and D of the study design. They are detailed in Tables 1–3 together with sources of data and measurement methods or technology.

**Table 1. Anamnestic and lifestyle variables.**

| variable | description | instrumentation | protocol/tool | type of variable |
|---|---|---|---|---|
| age | biological biomarker | ISS online platform | questionnaire | numerical |
| sex | biological biomarker | | questionnaire | categorical |
| ethnicity | biological biomarker | | questionnaire | categorical |
| living context and mobility | lifestyle variable | | questionnaire | categorical |
| nutrition | lifestyle variable accounting for average calories assumption/week | | QMV validated questionnaire [58] | numerical |
| hydration | lifestyle variable accounting for average liquid assumption/day | | questionnaire | numerical |
| smoking | lifestyle variable | | questionnaire | categorical |
| physical activity (hours) | lifestyle variable accounting for number of hours/week | | questionnaire | numerical |
| physical activity (description) | lifestyle variables accounting for all physical activity or training variables (environment included) | | questionnaire | descriptive (categorical) |
| footwear for physical activity | lifestyle variable | | questionnaire | categorical |
| footwear for working | lifestyle variable | | questionnaire | categorical |
| footwear for leisure | lifestyle variable | | questionnaire | categorical |
| working (description) | lifestyle variables delivering a structured description of working features (environment included) | | questionnaire | descriptive (categorical) |
| pathologies (other than overuse injuries within 2 years) | set of clinical variables | | questionnaire | descriptive (narrative and categorical) |
| overuse injuries within 2 years | set of clinical variables | | questionnaire | descriptive (narrative and categorical) |
| self-reported anthropometry | set of biological variables accounting for linear measurements of body segments | | questionnaire; ISS tutorial | numerical and multimedial (images) |

**Table 2. Biomechanical, physiological and environmental variables.**

| variable | description | instrumentation | protocol | type of variable |
|---|---|---|---|---|
| *physiological measurements* | | | | |
| body impedentiometry | biological markers of body composition* | professional Body Composition Scales relying on body impedance analysis technology (Tanita Europe BV; medical accuracy) | ISS lab procedure (acquisition before and after the functional test) | numerical and categorical |
| FEV1 (forced expiratory volume in 1 second) | functional biomarker of respiratory function | wireless spirometer (Air Next, Nuvoair, Boston, MA, U.S.; certified medical device) | ISS lab procedure (acquisition before the functional test) | numerical |
| tissue AGEs (advanced glycation end-products) | functional biomarker (used for estimation for cardiovascular risk) | professional tissue AGE reader (AGE Reader, Diagnoptics Technologies B.V., The Netherlands; certified medical device) | ISS lab procedure (acquisition before the functional test) | numerical |
| heart rate (HR) | functional biomarker of cardiac function | wireless band heart rate monitor (Polar H9, Polar Electro Italia; not a medical device) | ISS lab procedure (monitoring before, during and after the functional test) | numerical and categorical |
| oximetry | functional biomarker of oxygen saturation | finger oximeter (OXY-50, GIMA, Italy; certified medical device) | ISS lab procedure (acquisition before and after the functional test) | numerical |
| body temperature | physiological marker of thermoregulation | professional contactless thermometer (Visiofocus Pro, Tecnimed, Italy; certified medical device) | ISS lab procedure (acquisition before and after the functional test) | numerical |
| relative humidity–body surface | physiological marker of thermoregulation | professional wireless humidity meter (Inkbird IBS-TH1, Itech Europe Limited, UK; not a medical device) | ISS lab procedure (monitoring during the functional test) | numerical |
| body hydration | physiological marker of thermoregulation | professional instrument for body composition and hydration measurement (BCM—Body Composition Monitor, Fresenius Medical Care; certified medical device) | ISS lab procedure (acquisition before and after the functional test) | numerical |
| *biomechanical measurements* | | | | |
| temporal variables | functional markers of time process (temporal phases) | accurate optical portable instrumentation for gait analysis (solid with passive treadmill; OPTOGAIT, Microgate Italia, Italy; not a medical device) | ISS lab procedures (acquisition during the functional test) | numerical |
| external force | functional marker of individual foot loading pattern (ground reaction force) | accurate capacitive wearable (wireless) insoles (Loadsol, Novel, Germany; under certification as a medical device) | | |
| foot-ankle kinematics | functional marker of individual ankle joint kinematics (angular excursion in sagittal and frontal plane) | accurate bi-dimensional wearable (wireless) goniometers (Biometrics Ltd, UK; not a medical device) | | |
| lower leg electromyography (EMG) | functional marker of individual lower leg muscle function§ | accurate wearable (wireless) surface EMG (OT Bioelettronica, Italy; not a medical device) | | |
| legs and trunk accelerometry and inclinometry | functional marker of individual lower leg and of trunk motion (acceleration and orientation) | accurate wearable (wireless) inertial system (Opal, APDM, Portland, OR, U.S.; not a medical device) | | |
| functional test metrics | performance variables | onboard sensors of the passive treadmill (Power Mag, Toorx, Garlando S.p.A., Italy; certified to be used in professional sport centers) | | |
| fatigue perception and monitoring | functional marker of perceived and monitored fatigue | video recording of voice feedback based on Borg Scale (LG C922); HR monitoring | | |
| *environmental measurements* | | | | |
| temperature and relative humidity | environmental markers | accurate lab instrumentation (Testo 625 thermohygrometer with triple probe 0635 1540) | ISS lab procedures (monitoring during the functional test) | numerical |
| noise | environmental marker | accurate lab instrumentation (Chauvin-Arnoux-CDA-830 Sound Level Meter) | | |
| lighting | environmental marker | accurate lab instrumentation (photometer DeltaOhm HD2302.01, probe LP471 phot) | | |

* (body mass; body fat percentage; visceral fat; body water percentage; muscle mass; muscle quality; physique rating; bone density; basal metabolic rate; metabolic age)

§ surface electromyography (sEMG) of Tibialis Anterior and Lateral Gastrocnemius; ç time, distance, velocity and calories associated with the use of the passive treadmill.

**Table 3. Clinical and diagnostic variables.**

| variable | description | instrumentation | protocol | type of variable |
|---|---|---|---|---|
| complete blood count | biomarker of general health status | qualified chemical lab instrumentation | qualified chemical lab procedures | numerical |
| glycemia | biomarker of general health status | qualified chemical lab instrumentation | qualified chemical lab procedures | numerical |
| Glycosylated hemoglobin (HbA1c) | biomarker of prolonged hyperglicaemia | qualified chemical lab instrumentation | qualified chemical lab procedures | numerical |
| triglycerides | biomarker of general health status | qualified chemical lab instrumentation | qualified chemical lab procedures | numerical |
| total cholesterol | biomarker of general health status | qualified chemical lab instrumentation | qualified chemical lab procedures | numerical |
| HDL cholesterol, | biomarker of general health status | qualified chemical lab instrumentation | qualified chemical lab procedures | numerical |
| azotemia | biomarker of general health status | qualified chemical lab instrumentation | qualified chemical lab procedures | numerical |
| creatinine | biomarker of general health status | qualified chemical lab instrumentation | qualified chemical lab procedures | numerical |
| calcium | biomarker of general health status | qualified chemical lab instrumentation | qualified chemical lab procedures | numerical |
| iron | biomarker of general health status | qualified chemical lab instrumentation | qualified chemical lab procedures | numerical |
| ferritin | biomarker of general health status | qualified chemical lab instrumentation | qualified chemical lab procedures | numerical |
| urinalysis | biomarker of general health status | qualified chemical lab instrumentation | qualified chemical lab procedures | descriptive (categorical) |
| 25(OH)D level | biomarker of serum 25-Hydroxyvitamin D accounting for Vitamin D3 level and possible deficiency | qualified chemical lab instrumentation relying on chemiluminescent technology | qualified chemical lab procedures; Abbott procedures and reagents | numerical |
| bone mineral density | biomarkers of bone mineral density status and risk of fracture* | qualified imaging/diagnostic lab instrumentation (last generation dual-energy x-ray absorptiometry, (DEXA)) | qualified imaging/diagnostic lab procedures | numerical |

* T-score (bone mass possessed in comparison with young adults of the same sex); Z-score (bone mass in comparison to other people of the same sex and age).

- Variables associated with the collection of *anamnestic and lifestyle data* (Table 1). These variables have been selected by focussing on the main objectives of the study and on the basis of the systematic review of the literature, which helped defining the possible biological, clinical, physiological and lifestyle-related risk factors [38–49]. For those variables classified as "narrative", a Likert-based approach will be used to quantitatively score them [54].

- Variables associated with the *instrumental functional assessment* (Table 2).

  ○ The core of the instrumental functional assessment is the performance of an adapted version of the six minutes walking test (6MWT) [55]: the participant will be asked to walk on a passive treadmill, at 0 slope, "as fast as possible without running and trying to maintain the same pace during the test". Moreover, every 60 seconds, the participant will be questioned to self-report the perceived fatigue by an increasing number from 6 to 20 (Borg scale [56]). At the end of the 6MWT, if heart rate monitor and perceived fatigue allow it, the 6MWT will be repeated at increased slope (level 4 of 9). During the test, relevant biomechanical variables (kinematics, kinetics, EMG) will be monitored, each one acquired at pre-established sampling rate. Most variables in this group are quantitative, continuous variables, with the only exception of the Borg scale.

○ Before performing the functional test, a set of anthropometric and physiological variables will be measured, mostly quantitative and continuous. Oxygen saturation will be measured again 30s after the test.

○ During the test, environmental variables will be monitored as well.

○ Participants and operators will keep the EU Standard FFP2 facial mask (U.S./China Standard N95/KN95) during the entire data acquisition session, 6MWTs included [57].

○ All devices included in the toolkit had been technically characterized and assessed during the first part of the MOVIDA project (implementation of the toolkit) to check the appropriateness of their measurement performance; some of them are certified as medical devices (as indicated in Table 2, column 3); for those which are not, or not yet, technical documentation from the Companies has been checked to guarantee for their safe use on human volunteers.

- Variables associated with the specific *clinical-diagnostic examination*. Blood sampling and chemical lab analysis will allow to acquire routine clinical biomarkers related to general health status and to assess the Vitamin D level through the evaluation of 25(OH)D level [23]. Additionally, the DEXA diagnostic examination for bone densitometry will deliver the associated biomarkers in terms of T-score and Z-score (Table 3).

## Bias

**Collection of anamnestic and lifestyle data.** Validated questionnaires were used whenever possible. Ad-hoc questionnaires were double-checked and tested by the Project team. Score stratification will be performed on a Likert-based approach by a couple of researchers of the Project team and discussion will be planned in case of disagreement.

**Collection of data from the instrumental functional assessment.** Biomechanical measurements will rely on accurate measurement instrumentation previously validated at the Authors' Labs and periodically tested. All devices purposely purchased to monitor physiological and environmental parameters were validated with respect to reference Lab instrumentation, and systematic bias purposely modelled. To minimise bias in self-reporting on perceived fatigue during the 6MWT, the Borg scale (score 6–20) was selected rather than the traditional 1–10 Visual Analogue Scale (VAS).

**Variables associated with the specific clinical-diagnostic examination.** The chemical lab of the Celio Army Medical Center will be in charge for all participants blood sampling and analysis. Evaluation of 25(OH)D will be performed through validated instrumentation and procedure (ARCHITECT i2000, Abbott). Bone mineral density will be evaluated by a Qualified Diagnostic Lab (the same for all participants) equipped with last generation instrumentation for dual-energy x-ray absorptiometry (DEXA).

## Study size

A total of 150 participants will be examined during the study. The sample size estimation was performed by using the G*Power tool (version 3.1.9.7, https://www.psychologie.hhu.de/en/). An a-priori estimation of required sample size was asked for a statistical test ANCOVA with fixed effects, main effects and interactions. Effect size was established at 0.4 based on expected variations of kinetic and kinematic parameters during the 6MWT; $\alpha$ error was fixed at 5%; power was fixed at 80%. For the 3 groups of participants and 2 covariates (sex and military or civilian condition) the estimated total sample size was 146.

### Quantitative variables

**Anamnestic and lifestyle data.** Quantitative information delivered by participants through the online platform will be double-checked for congruity before importing them into the study database. Qualitative information will be checked and stratified using a Likert-based, 5 points approach. Eventual feedback or clarification will be asked to participants for any unclear information before feeding the database.

**Instrumental functional assessment data.** Biomechanical parameters will be acquired at the most adequate sampling rate (Table 2), filtered to cut off high frequency noise, processed to extrapolate instantaneous quantities (i.e. peak values), global quantities (i.e. impulses integrated over specified observation windows), temporal patterns (i.e. force curves) or frequency spectrum (i.e. amplitude frequency spectrum of EMG signal). Physiological and environmental quantities will be treated as instantaneous numerical variables.

**Clinical-diagnostic data.** All numerical values will be delivered after a quality check at the processing Lab and will directly feed the study database. An additional check will only be performed at ISS to guarantee for completeness and correctness or import and storage process.

### Statistical methods

According to amount and distribution of missing values, the most effective strategy will be chosen to cope with this issue, ranging from deleting the participant from the study (likely the very last option to choose, due to the limited number of participants), deleting a variable (possibly acceptable if the variable is redundant of proxy of another variable), imputing mean or median values, imputing values on Bayesian-based procedures. A machine learning approach might be also explored to predict missing values.

The t-test, one and two-way ANOVA with post-hoc corrections, and ANCOVA tests will be used as consolidated statistical approaches to investigate relevant differences among the groups with respect to biomechanical parameters, and for all those quantities whose distribution will satisfy the mandatory requirements of those statistical tools. Non-parametric statistics will be rather used for non-normal continuous variables. A chi-squared test will be used for quantitative discrete variables. Logistic regression models and generalized linear models will be used to account for risk and confounding factors. The feasibility and relevance of cluster analysis will be explored as well. Last, ROC curves will be tentatively used to characterize the sensitivity/specificity trade-offs for possibly detected binary risk classifiers.

Data processing, analysis and statistics will be performed by ad-hoc codes and procedures developed and implemented in Matlab and R environments. The database will be structured and managed in R as well.

### Ethics and dissemination

The study received the approval of the Celio Army Medical Center Ethical Committee (Prot. N. CE/2020u/19/a-11/07/2020), and the approval of the Data Protection Officer (DPO) of Italian National Institute of Health for the compliance with EU General Data Protection Regulation 2016/679. Each volunteer will sign the informed consent before data collection. Formal agreement has been obtained by the MOVIDA Project funder (Italian Ministry of Defence) for publication of the MOVIDA study protocol. Formal agreement will also be asked for dissemination of MOVIDA study results.

## Discussion

The toolkit relies on a wide and comprehensive set of heterogeneous parameters exploring several dimensions correlated to risk of overuse-related injuries.

The clinical and diagnostic examination will rely on validated and certified procedures of the qualified chemical and diagnostic Laboratory of the Celio Army Medical Center.

The biomechanical assessment protocol makes use of several instruments and may be fairly complex to conduce. Validation phase showed that a minimum of two experimenters is mandatory to reliably conduct the experiment and acquire or record all variables and data. Instruments synchronization is fundamental for the accurate signal processing chain thus the principle of redundancy is applied: namely, trigger signals, mechanical and audio trigger events, devices clock synchronization and multiple high-speed videos recording will be used for the purpose. This complexity, however, is counterbalanced by two major advantages: it allows to optimise the participant's time burden, since all variables are acquired during the execution of the instrumented 6MWT; it helps investigating reliability, redundancy and relevance of several biomechanical parameters associated with the test. All instruments used during this phase of the study have been characterized with respect to reference laboratory instrumentation.

Despite the study protocol deals with young healthy adults only, it has been designed and is expected to be safely applicable to individuals with different level of physical ability and to the elderly.

The cross-sectional study alone, though helpful to correlate changes in biomechanical parameters due to fatigue with fitness status and physical training, will not allow to model overuse risk predictors; it will however allow to set the stage for further developments that can be obtained with follow up.

At a more general level, the overall execution of the project might cope with some possible limitations. First of all, it must be noted that Sars-Cov-2 pandemic situation not only delayed all data collection phases; it also contributed, and is currently contributing, to partial rescheduling of professional and elite sport events which are either suspended, postponed or cancelled, thus resulting in poor availability of enrolled volunteers in the period June-August 2021. In addition, the hot temperatures and the concomitance with the "working summer break" in Italy will further slowdown the experimental phase timeline. As for the strengths and relevance of the overall project, some points are worth to be here highlighted. The toolkit integrates, in a single assessment, several preventive perspectives on the risk of injury, allowing their combined monitoring. Although currently performed with selected, well characterized devices, the protocol is based on off-the-shelf sensors and allows for an easy up-date with evolving sensor market or for being implemented by other research groups with different sensors, provided that quality of the used sensors is similarly assessed. At this stage of the project we want to check the suitability of the protocol in a controlled situation such as the laboratory environment; nevertheless, the whole experimental setup can be moved outside the lab and used in ecological conditions, more similar to those of real-life. Last, despite this project deals with an observational study only, the developed toolkit may effectively serve as support for interventional studies to reduce injury risk, as is might be the case for Vitamin D supplementation studies.

## Supporting information

**S1 File. STROBE statement—checklist of items that should be included in reports of cross-sectional studies.**
(DOCX)

## Author Contributions

**Conceptualization:** Valeria Belluscio, Amaranta S. Orejel Bustos, Valentina Camomilla, Claudia Giacomozzi.

**Funding acquisition:** Claudia Giacomozzi.

**Investigation:** Claudia Giacomozzi.

**Methodology:** Valeria Belluscio, Amaranta S. Orejel Bustos, Valentina Camomilla, Claudia Giacomozzi.

**Project administration:** Claudia Giacomozzi.

**Resources:** Claudia Giacomozzi.

**Supervision:** Valentina Camomilla, Claudia Giacomozzi.

**Validation:** Valeria Belluscio, Amaranta S. Orejel Bustos, Valentina Camomilla, Francesco Martelli, Claudia Giacomozzi.

**Visualization:** Valeria Belluscio, Amaranta S. Orejel Bustos, Valentina Camomilla, Claudia Giacomozzi.

**Writing – original draft:** Valeria Belluscio, Amaranta S. Orejel Bustos, Valentina Camomilla, Claudia Giacomozzi.

**Writing – review & editing:** Valeria Belluscio, Amaranta S. Orejel Bustos, Valentina Camomilla, Francesco Rizzo, Tommaso Sciarra, Marco Gabbianelli, Raffaella Guerriero, Ornella Morsilli, Francesco Martelli, Claudia Giacomozzi.

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
