## [Decision Letter · Decision Letter 0]

16 Jun 2021

PONE-D-21-06273

Experimental study protocol of the project “MOtor function and VItamin D: toolkit for motor performance and risk Assessment (MOVIDA)”

PLOS ONE

Dear Dr. Giacomozzi,

Thank you for submitting your manuscript to PLOS ONE. After careful consideration, we feel that it has merit but does not fully meet PLOS ONE’s publication criteria as it currently stands. Therefore, we invite you to submit a revised version of the manuscript that addresses the points raised during the review process.

We look forward to receiving your revised manuscript.

Kind regards,

Georg Osterhoff, M.D.

Academic Editor

PLOS ONE

Additional Editor Comments (if provided):

Reviewers' comments:

Reviewer's Responses to Questions

**Comments to the Author**

1. Does the manuscript provide a valid rationale for the proposed study, with clearly identified and justified research questions?

Reviewer #1: Partly

Reviewer #2: Yes

2. Is the protocol technically sound and planned in a manner that will lead to a meaningful outcome and allow testing the stated hypotheses?

Reviewer #1: Yes

Reviewer #2: Yes

3. Is the methodology feasible and described in sufficient detail to allow the work to be replicable?

Reviewer #1: Yes

Reviewer #2: No

4. Have the authors described where all data underlying the findings will be made available when the study is complete?

Reviewer #1: No

Reviewer #2: No

5. Is the manuscript presented in an intelligible fashion and written in standard English?

Reviewer #1: Yes

Reviewer #2: Yes

6. Review Comments to the Author

You may also provide optional suggestions and comments to authors that they might find helpful in planning their study.

Reviewer #1: 1-Please check the spelling of vitamin D and correct it all over the paper; vitamin D or vit D?

2-The rationale of evaluation vitamin D is not clear; please use reference for the prevalence of vitamin D deficiency among military staff and soldiers. Also, please explain why you do not start vitamin D supplementation (even 50000 IU per month) to prevent hypovitaminosis among military staff. Is it not cost effective?

3-Line 171. Please add the reference for WHO classification.

4- Any potential limitations regarding the project? Please add to the discussion. Also you can mention the strength points of this work.

Reviewer #2: This study protocol is clearly described, the rationale and motivations well-identified and the manuscript is clearly written and easy to read.

Although I think the manuscript is ready for publication, I saw a few unclear aspects and I would like to see them solved if possible:

- which devices will be used for the physiological and biomechanical parameters

- are they all medical devices?

- will biomechanical measurements be performed also in ecological conditions (not in the lab)?

- will fatigue be assessed through subjective reporting only?

Minor: on line 263 there is an undefined symbol

7. PLOS authors have the option to publish the peer review history of their article (what does this mean?). If published, this will include your full peer review and any attached files.

Reviewer #1: No

Reviewer #2: No

---

## [Author Response · Author response to Decision Letter 0]

2 Jul 2021

We thank the Reviewers for the constructive comments. Below, we reported how we have addressed each of the Reviewers’ comments. Corresponding changes have been highlighted in the manuscript.

Reviewer #1: 

1-Please check the spelling of vitamin D and correct it all over the paper; vitamin D or vit D? 

Authors. We checked and corrected the incongruencies adopting “Vitamin D” throughout the Manuscript.

2-The rationale of evaluation vitamin D is not clear; please use reference for the prevalence of vitamin D deficiency among military staff and soldiers. Also, please explain why you do not start vitamin D supplementation (even 50000 IU per month) to prevent hypovitaminosis among military staff. Is it not cost effective? 

Authors.

Three answers have been formulated to address this issue.

Answer 1. Despite its contribution to altered motor function performance is still partly debated, more and more scientific evidence is reported in the literature which links D hypovitaminosis with stress fractures/injuries, and a high level of interest is found in the literature with respect to the development of effective nutrition plans, at the level of global public health but also with respect to Service Members in the Army. In this study, we aim to monitor the Vitamin D biomarker within our cohort and to possibly include it as a co-variate in our generalized models. This concept has been better clarified in the objectives of the study through the adaptation of point ii of the objective list as follows: “ii) modelling of risk factors interaction, including modelling to associate the assessment outcomes with both retrospective (previous injuries occurred within 2 years) and prospective (during a follow-up period up to 2 years) information; in particular, the Vitamin D biomarker will be monitored within the examined cohort to possibly include it as a relevant co-variate in the models;”

Answer 2. The sentence at previous lines 70-72 has been completed with few examples of D hypovitaminosis prevalence, and 2 references have been added purposely, now listed as reference #28 and #29. The sentence has been re-written as follows: “Moreover, in the military populations, there is a significant predisposition to develop low levels of Vitamin D or to undergo relevant level fluctuations [7,9], with reported high D hypovitaminosis incidence and prevalence among soldiers [28-30]. This phenomenon has been partly ascribed to low exposure to sunlight [22] and to the lack of adequately supplemented food [7].”

Answer 3. Within the MOVIDA Project, the experimental study had been designed as an observational study, for several reasons but mainly due to the limited duration of the project itself, not enough long for a robust interventional study. However, the developed toolkit may effectively serve as support for interventional studies to reduce injury risk, also addressing D hypovitaminosis treatment. This “potentiality” has been added as a strength in the Discussion, while addressing comment #4.

3-Line 171. Please add the reference for WHO classification.

Authors. The following reference has been added as ref #53 “WHO guidelines on physical activity and sedentary behaviour. Geneva: World Health Organization; 2020. Licence: CC BY-NC-SA 3.0 IGO. ISBN 978-92-4-001512-8”. Successive references have been shifted accordingly. A refuse was also corrected in previous line 171, namely the lower limit “1” was replaced by “2.5” which is the proper boundary between sedentary and healthy active adults.

4- Any potential limitations regarding the project? Please add to the discussion. Also you can mention the strength points of this work. 

Authors. Following this constructive comment, limitations and strengths have been added to the Discussion. The following paragraph has been added: “At a more general level, the overall execution of the project might cope with some possible limitations. First of all, it must be noted that Sars-Cov-2 pandemic situation not only delayed all data collection phases; it also contributed, and is currently contributing, to partial rescheduling of professional and elite sport events which are either suspended, postponed or cancelled, thus resulting in poor availability of enrolled volunteers in the period June-August 2021. In addition, the hot temperatures and the concomitance with the “working summer break” in Italy will further slowdown the experimental phase timeline. As for the strengths and relevance of the overall project, some points are worth to be here highlighted. The toolkit integrates, in a single assessment, several preventive perspectives on the risk of injury, allowing their combined monitoring. Although currently performed with selected, well characterized devices, the protocol is based on off-the-shelf sensors and allows for an easy up-date with evolving sensor market or for being implemented by other research groups with different sensors, provided that quality of the used sensors is similarly assessed. At this stage of the project we want to check the suitability of the protocol in a controlled situation such as the laboratory environment; nevertheless, the whole experimental setup can be moved outside the lab and used in ecological conditions, more similar to those of real-life. Last, despite this project deals with an observational study only, the developed toolkit may effectively serve as support for interventional studies to reduce injury risk, as is might be the case for Vitamin D supplementation studies.”

Reviewer #2: 

This study protocol is clearly described, the rationale and motivations well-identified and the manuscript is clearly written and easy to read.

Authors. We want to thank the Reviewer for his/her appreciation and hope to have satisfactorily addressed all below comments

Although I think the manuscript is ready for publication, I saw a few unclear aspects and I would like to see them solved if possible:

- which devices will be used for the physiological and biomechanical parameters

Authors. Devices have been specified in Table 2, column 3

- are they all medical devices?

Authors. Certified medical devices had been previously indicated in Table 2, column 3. To better clarify the issue, details have been added (same column) for those devices which are not (or not yet) certified as medical devices. Additionally, a small paragraph has been added ahead of Table 2 to state that all used devices are safe for the use on human volunteers and that all of them have been previously characterized with respect to their technical performance. (“All devices included in the toolkit had been technically characterized and assessed during the first part of the MOVIDA project (implementation of the toolkit) to check the appropriateness of their measurement performance; some of them are certified as medical devices (as indicated in Table 2, column 3); for those which are not, or not yet, technical documentation from the Companies has been checked to guarantee for their safe use on human volunteers”)

- will biomechanical measurements be performed also in ecological conditions (not in the lab)? 

Authors. Thank you for the pertinent comment. At this stage of the project the toolkit is going to be used in lab, since one of our objectives is to check the suitability of the protocol in a controlled situation such as the laboratory environment. Nevertheless, the whole experimental setup can be moved “outside” and used in ecological conditions, more similar to those of real-life. We mentioned this potential strength in the Discussion.

- will fatigue be assessed through subjective reporting only?

Authors. Fatigue will be monitored and assessed at various levels. As previously reported in Table 2, perceived fatigue will be monitored by using the Borg’s Rating of Perceived Exertion Scale, which has been proved to be highly correlated with both Heart Rate and blood lactate concentration changes during exercise (Scherr et al. Associations between Borg’s rating of perceived exertion and physiological measures of exercise intensity, Eur J Appl Physiol. 2013). Additionally, fatigue will be monitored through Heart Rate continuous measurement (listed in Table 2 as physiological measurement). Last, muscle fatigue indicators will be estimated based on changes of all biomechanical measurements, especially those coming from sEMG. Line “fatigue perception” in Table 2 has been slightly modified for better clarity.

Minor: on line 263 there is an undefined symbol

Authors. The typo has been corrected.

---

## [Editor Report · Decision Letter 1]

6 Jul 2021

Experimental study protocol of the project “MOtor function and VItamin D: toolkit for motor performance and risk Assessment (MOVIDA)”

PONE-D-21-06273R1

Dear Dr. Giacomozzi,

We’re pleased to inform you that your manuscript has been judged scientifically suitable for publication and will be formally accepted for publication once it meets all outstanding technical requirements.

Kind regards,

Georg Osterhoff, M.D.

Academic Editor

PLOS ONE

Additional Editor Comments (optional):

The authors addressed all concerns raised by the Reviewers sufficiently.
---

## [Editor Report · Acceptance letter]

13 Jul 2021

PONE-D-21-06273R1 

Experimental study protocol of the project “MOtor function and VItamin D: toolkit for motor performance and risk Assessment (MOVIDA)” 

Dear Dr. Giacomozzi:

I'm pleased to inform you that your manuscript has been deemed suitable for publication in PLOS ONE. Congratulations! Your manuscript is now with our production department. 

Kind regards, 

on behalf of

Dr. Georg Osterhoff 

Academic Editor

PLOS ONE